# The kinetoplastid-infecting Bodo saltans virus (BsV), a window into the most abundant giant viruses in the sea

Christoph M Deeg[1], Cheryl-Emiliane T Chow[2], Curtis A Suttle[1,2,3,4]*

[1]Department of Microbiology and Immunology, University of British Columbia, Vancouver, Canada; [2]Department of Earth, Ocean and Atmospheric Sciences, University of British Columbia, Vancouver, Canada; [3]Department of Botany, University of British Columbia, Vancouver, Canada; [4]Institute for the Oceans and Fisheries, University of British Columbia, Vancouver, Canada

**Abstract** Giant viruses are ecologically important players in aquatic ecosystems that have challenged concepts of what constitutes a virus. Herein, we present the giant Bodo saltans virus (BsV), the first characterized representative of the most abundant group of giant viruses in ocean metagenomes, and the first isolate of a klosneuvirus, a subgroup of the *Mimiviridae* proposed from metagenomic data. BsV infects an ecologically important microzooplankton, the kinetoplastid *Bodo saltans*. Its 1.39 Mb genome encodes 1227 predicted ORFs, including a complex replication machinery. Yet, much of its translational apparatus has been lost, including all tRNAs. Essential genes are invaded by homing endonuclease-encoding self-splicing introns that may defend against competing viruses. Putative anti-host factors show extensive gene duplication via a genomic accordion indicating an ongoing evolutionary arms race and highlighting the rapid evolution and genomic plasticity that has led to genome gigantism and the enigma that is giant viruses.
DOI: https://doi.org/10.7554/eLife.33014.001

*For correspondence:
suttle@science.ubc.ca

Competing interests: The authors declare that no competing interests exist.

## Introduction

Viruses are the most abundant biological entities on the planet and there are typically millions of virus particles in each milliliter of marine or fresh waters that are estimated to kill about 20% of the living biomass each day in surface marine waters (*Suttle, 2007*). This has major consequences for global nutrient and carbon cycles, as well as for controlling the composition of the planktonic communities that are the base of aquatic foodwebs. Although the vast majority of aquatic viruses are less than 100 nm in diameter and primarily infect prokaryotes, it is increasingly clear that a subset of the viruses in aquatic ecosystems are comparative Leviathans that have been colloquially classified as giant viruses.

The first isolated giant virus in the family that later became known as the *Mimiviridae*, infects a marine heterotrophic flagellate that was initially identified as *Bodo* sp. (*Garza and Suttle, 1995*), and later shown to be *Cafeteria roenbergensis* (*Fischer et al., 2010*). Subsequently, the isolation and sequencing of mimivirus, a giant virus infecting *Acanthamoeba polyphaga* (*La Scola et al., 2003*; *Raoult et al., 2004*), transformed our appreciation of the biological and evolutionary novelty of giant viruses. This led to an explosion in the isolation of different groups of giant viruses infecting *Acanthamoeba* spp. including members of the genera *Pandoravirus*, *Pithovirus*, *Mollivirus*, *Mimivirus* and *Marseillevirus* (*Boughalmi et al., 2013*; *Colson et al., 2013*; *Legendre et al., 2014*; *Legendre et al., 2015*; *Philippe et al., 2013*). Although each of these isolates expanded our understanding of the evolutionary history and biological complexity of giant viruses, all are pathogens of *Acanthamoeba* spp., a widespread taxon that is representative of a single evolutionary branch of

**eLife digest** In oceans, rivers and lakes, there are about a million viruses in every milliliter of water. Most of these viruses are tiny, often 10 or 100 times smaller than bacteria. However, a few reach a similar size and complexity to bacteria, and so stand out as relative giants.

Relative to other viruses, Giant Viruses have much more DNA in their genome, which in turn provides the genetic template to produce the proteins that allow viruses to reproduce largely independently of its host. Typically, more than half of the genes encoded by Giant Viruses have no evident similarity to genes from other viruses or cellular life. Sequencing DNA from ocean water suggests that Giant Viruses are abundant and ecologically important; yet, few have been isolated from the microbes that they infect. Without being able to study Giant Viruses in the laboratory, little can be known about their biology, the way they infect their hosts, and their broader influence on aquatic life.

Deeg et al. have now isolated and characterized the giant Bodo saltans virus (BsV), a Giant Virus that infects an ecologically important microbe commonly found in aquatic environments. Sequencing the genome of BsV revealed many previously unknown genes, as well as several unusual features. For example, the genome contains movable genetic elements that might help to fend off other giant viruses by cutting their genomes. In addition, the set of genes used by BsV to translate mRNA templates into proteins differs from those found in other giant viruses, implying that they are not derived from a more complex common ancestor.

The size of the genome appears to have grown rapidly by the duplication of genes at the end of the genome – a feature known as a genomic accordion. The identity of the duplicated genes suggests that there is an evolutionary arms race with its host that forces genome expansion. Further studies of the BsV genome could help researchers to understand the origin of gigantism in the genomes of giant viruses.

DOI: https://doi.org/10.7554/eLife.33014.002

eukaryotes, and which is not a major component in the planktonic communities that dominate the world's oceans and large lakes.

As knowledge of mimiviruses infecting *Acanthamoeba* spp. has expanded, it has become evident based on analysis of metagenomic data that giant viruses and their relatives are widespread and abundant in aquatic systems (*Hingamp et al., 2013*; *Mozar and Claverie, 2014*; *Schulz et al., 2017*). However, except for Cafeteria roenbergensis Virus (CroV) that infects a microzooplankton (*Fischer et al., 2010*), and the smaller phytoplankton-infecting viruses Phaeocystis globosa virus PgV-16T (*Santini et al., 2013*), Chrysochromulina Ericina Virus (*Gallot-Lavallée et al., 2017*), and Aureococcus anophagefferens virus (*Moniruzzaman et al., 2014*), the only members of the *Mimiviridae* that have been isolated and characterized infect *Acanthamoeba* spp.

Motivated by the lack of ecologically relevant giant-virus isolates, we isolated and screened representative microzooplankton in order to isolate new giant-viruses that can serve as model systems for exploring their biology and function in aquatic ecosystems. Herein, we present Bodo saltans virus (BsV), a giant virus that infects the ecologically important kinetoplastid microzooplankter *Bodo saltans*, a member of the phylum Euglenazoa within the supergroup Excavata. This group of protists is well represented by bodonids in freshwater environments and by diplonemids in the oceans (*Flegontova et al., 2016*; *Simpson et al., 2006*). Kinetoplastids are remarkable for their highly unusual RNA editing and having a single large mitochondrion, the kinetoplast, that contains circular concatenated DNA (kDNA) that comprises up to 25% of the total genomic content of the cell (*Shapiro and Englund, 1995*; *Simpson et al., 2006*), and are well known as causative agents of disease in humans (e.g. Leishmaniasis and sleeping sickness) and livestock (*Jackson et al., 2016*; *Mukherjee et al., 2015*). At 1.39 MB, BsV has one of the largest described complete genome of a cultured strain within the giant virus family *Mimiviridae*. Based on a recruitment analysis of metagenomic reads, BsV is representative of the most abundant group within the *Mimiviridae* in the ocean and is the only isolate of the klosneuviruses, a group only known from metagenomic data (*Schulz et al., 2017*). The BsV genome exhibits evidence of significant genome rearrangements and recent adaptations to its host.

## Results

### Isolation and infection kinetics

In an effort to isolate giant viruses that infect ecologically relevant organisms, we isolated protistan microzooplankton from a variety of habitats and screened them against their associated virus assemblages. One such screen using water collected from a temperate eutrophic pond in southern British Columbia, Canada, yielded a giant virus that we have classified as Bodo saltans virus, Strain NG1 (BsV-NG1) that infects an isolate of the widely occurring kinetoplastid, *Bodo saltans* (Strain NG, CCCM6296). The addition of BsV to a culture of *Bodo saltans* ($\sim2.5 \times 10^5$ cells ml$^{-1}$) at a virus particle to cell ratio of two, measured by flow cytometry, resulted in free virus particles 18 hr later. Viral concentrations peaked at $2.5 \times 10^7$ particles ml$^{-1}$, while host cell density dropped to 25% of uninfected control cultures (*Figure 1*). The closely related strain *Bodo saltans* HFCC12 could not be infected by BsV-NG1, suggesting strain specificity.

### Virus morphology and replication kinetics

Transmission electron microscopy (TEM) revealed that BsV is an icosahedral particle approximately 300 nm in diameter (*Figure 2A*). The particle consists of at least six layers akin to observations of Acanthamoeba polyphaga mimivirus (ApMV) (*Mutsafi et al., 2013*). The DNA-containing core of the virion was surrounded by a core wall and an inner membrane, and a putative membrane sitting under a double-capsid layer (*Figure 2A*). A halo of approximately 25 nm surrounds the virion. A possible stargate-like structure, as observed in ApMV, is associated with a depression of the virus core *Figure 2—figure supplement 1B,D*), which presumably releases the core from the capsid during infection (*Klose et al., 2010*; *Mutsafi et al., 2014*).

The healthy *Bodo saltans* cell presents intricate intracellular structures, including the characteristic kinetoplast and a pronounced cytostome and cytopharynx (*Figure 2B*, *Figure 2—figure supplement 1A*). In infected cells, virus factories were always observed in the cell's posterior and particles always matured toward the posterior cell pole in a more spatially organized way compared to other *Mimiviridae* (*Figure 2C*, *Figure 2—figure supplement 1C*) (*Mutsafi et al., 2010*). As infection progressed, the Golgi apparatus disappeared and the nucleus degraded, as evidenced by the loss of the nucleolus and heterochromatin (*Figure 2B,C*); yet, the kinetoplast remained intact, as indicated by the persistence of the characteristic kDNA structure (*Figure 2B,C*, *Figure 2—figure supplement 1A,C*). Virus factories were first observed at 6 hr post-infection (p.i.) as electron-dense diffuse areas in the cytoplasm. By 12 h p.i., the virus factory had expanded significantly and reached a maximum size of about one-third of the host cell, taking up most of the cytoplasm. The first capsid structures appeared at this time. At 18 h p.i., the first mature virus particles were observed, coinciding with the first free virus particles observed by flow cytometry (*Figure 1*). By 24 h p.i., most infected cells were at the late stage of infection with mature virus factories (*Figure 2C,D*). During virus replication,

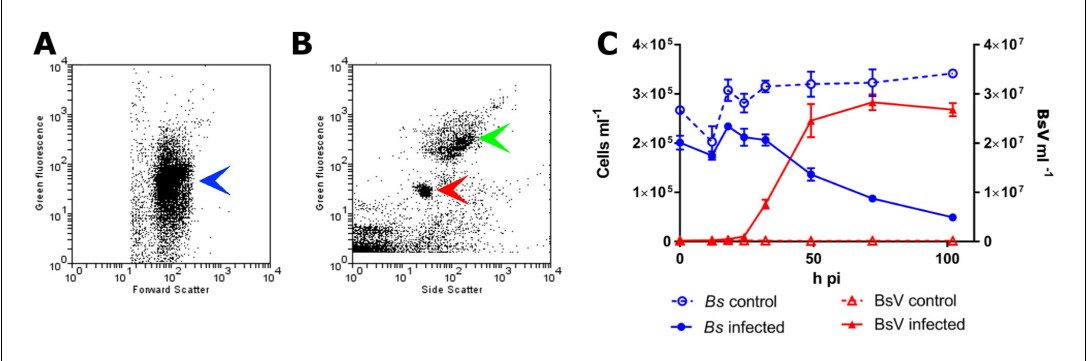

**Figure 1.** BsV induced lysis observed by flow cytometry. (A) Flow cytometry profile of uninfected Bodo saltans stained with Lysotracker (blue arrow head) (B) Flow cytometry profile of BsV (red arrow head) and bacteria (green arrow head) stained with SYBR Green. (C) The abundances of B. saltans cells and BsV particles after infection at a particle to cell ratio of 2.
DOI: https://doi.org/10.7554/eLife.33014.003

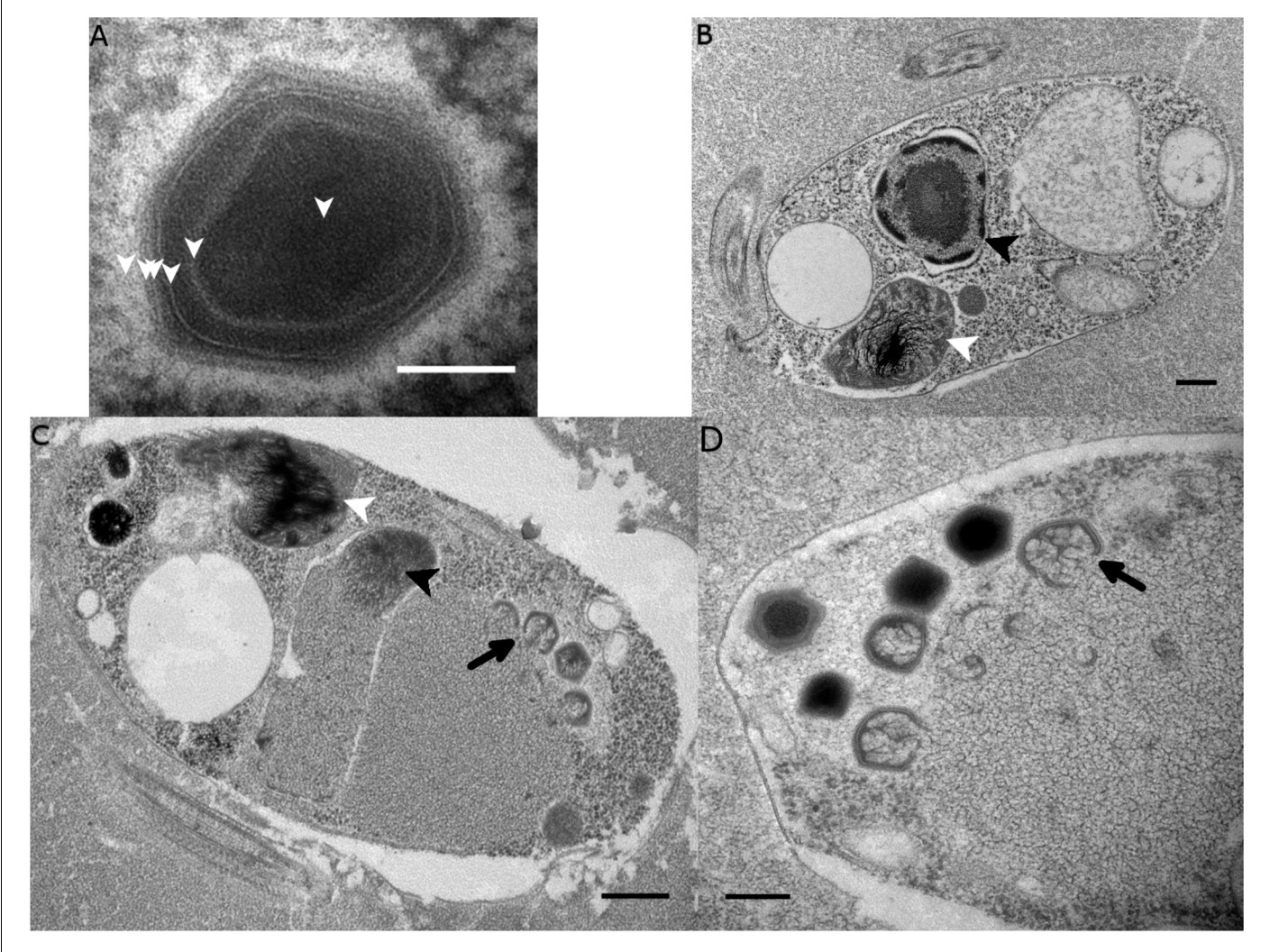

**Figure 2.** Ultrastructure of BsV particles and replication. (**A**) Mature BsV virion: DNA containing core is surrounded by two putative membranous layers. The capsid consists of at least two proteinaceous layers. The bright halo hints to the presence of short (~40 nm) fibers as observed in ApMV. The top vertex of the virion contains a possible stargate structure (See also *Figure 2—figure supplement 1*). (Scale bar = 100 nm) (**B**) Healthy *Bodo saltans cell*: Nucleus with nucleolus and heterochromatin structures (Back arrow head) and kinetoplast genome (white arrow head) are clearly visible. (Scale bar = 500 nm) (**C**) Cell of *Bodo saltans* 24 hr post-BsV infection: Most subcellular compartments of healthy cells have been displaced by the virus factory now taking up a third of the cell. Virion production is directed toward the periphery of the cell (black arrow). Kinetoplast genome remains intact (white arrow head) while the nuclear genome is degraded (black arrow head; Scale bar = 500 nm) (**D**) BsV virion assembly and maturation: Lipid vesicles migrate through the virion factory where capsid proteins attach for the proteinaceous shell. Vesicles burst and accumulate at the virus factory periphery where the capsid assembly completes (black arrow). Once the capsid is assembled, the virion is filled with the genome and detaches from the virus factory. Internal structures develop inside the virion in the cell's periphery where mature virions accumulate until the host cell bursts (Scale bar = 500 nm). See *Figure 2—figure supplement 1* for further information.

DOI: https://doi.org/10.7554/eLife.33014.004
The following figure supplement is available for figure 2:

**Figure supplement 1.** Ultrastructure of BsV particles and replication.
DOI: https://doi.org/10.7554/eLife.33014.005

membrane vesicles were recruited through the virus factory where capsid proteins accumulated and disrupted the vesicles (*Figure 2D*) (*Mutsafi et al., 2013*). The vesicle/capsid structures accumulated in the periphery of the virus factory where the capsid was formed (*Figure 2C,D*). Once the capsid was completed, the viral genome was packaged into the capsid at the vortex opposite to the putative stargate structure (*Figure 2C,D*). The internal structures of the virus particle matured in the cell

periphery and accumulated below the host cytoplasmic membrane where they often remained for an extended period of time (*Figure 2D*, *Figure 2—figure supplement 1D*). Besides being released during cell lysis, mature virus particles were observed budding in vesicles from the host membrane, reminiscent of a mechanism described for *Marseillevirus* (*Figure 2—figure supplement 1D*) (*Arantes et al., 2016*).

## Genome organization

Combined PacBio RSII and Illumina MiSeq sequencing resulted in the assembly of a 1,385,869 bp linear double-stranded DNA genome (accession number MF782455), making the BsV genome one of the largest complete viral genomes described to date, surpassing those of mimiviruses infecting *Acanthamoeba* spp. The GC content is 25.3% (*Figure 3*) and, as reported for other giant viruses (*Raoult et al., 2004*), much lower than the ~50% observed for *Bodo spp.* (*Jackson et al., 2016*); this suggests the absence of large scale horizontal gene transfer with the host in recent evolutionary history. The genome encodes 1227 predicted open-reading frames (ORFs) with a coding density of 85%, with the ORFs distributed roughly equally between the two strands consistent with the constant GC-skew (*Figure 3*). Unlike ApMV, BsV does not display a central peak in GC skew and therefore does not have an organized bacterial like origin of replication (*Raoult et al., 2004*). The genomic periphery has a slightly skewed GC ratio due to the tandem orientation of repeated ORFs. Codon preference is highly biased toward A/T-rich codons and the amino acids Lysine, Asparagine, Isoleucine, and Leucine (10, 9.8, 9.6, 8%), which are preferentially encoded by A/T only triplets. The translation of the predicted ORFs resulted in proteins ranging from 43 to 4840 aa in length with an average length of 320 aa. Promotor analysis revealed a highly conserved early promotor motif 'AAAAATTGA' that is identical to that found in mimiviruses and CroV (*Fischer et al., 2010*; *Priet et al., 2015*). A poorly conserved late promotor motif 'TGCG' surrounded by AT-rich regions was also observed. ORFs are followed by palindromic sequences, suggesting a hairpin-based transcription termination mechanism similar to ApMV (*Byrne et al., 2009*). Several non-coding stretches

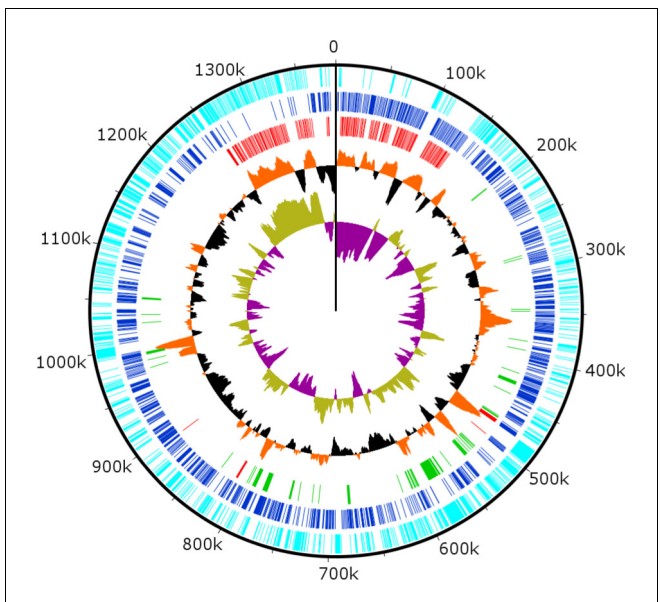

**Figure 3.** Circularized genome plot of the linear BsV genome. Circles from inside out: Olive/Purple: GC-skew; Orange/Black: % GC content plotted around average of 25.3%; Red: Ankyrin repeat domain-containing proteins; Green: Essential NCLDV conserved genes; Dark blue: Plus strand encoded ORFs; Light blue: Negative strand encoded ORFs. See *Figure 3—figure supplement 1* for further information on repeat regions.
DOI: https://doi.org/10.7554/eLife.33014.006
The following figure supplement is available for figure 3:

**Figure supplement 1.** Ankyrin repeat domain-containing proteins.
DOI: https://doi.org/10.7554/eLife.33014.007

rich in repetitive sequences were observed, but no function could be attributed to them. Based on a BLASTp analysis, 40% of ORFs had no significant similarity to any other sequences and remained ORFans (*Figure 4B*). Most proteins (27%) matched sequences from eukaryotes; two % of these matched best to sequences from isolates of *B. saltans*. The next largest fraction (22%) were most

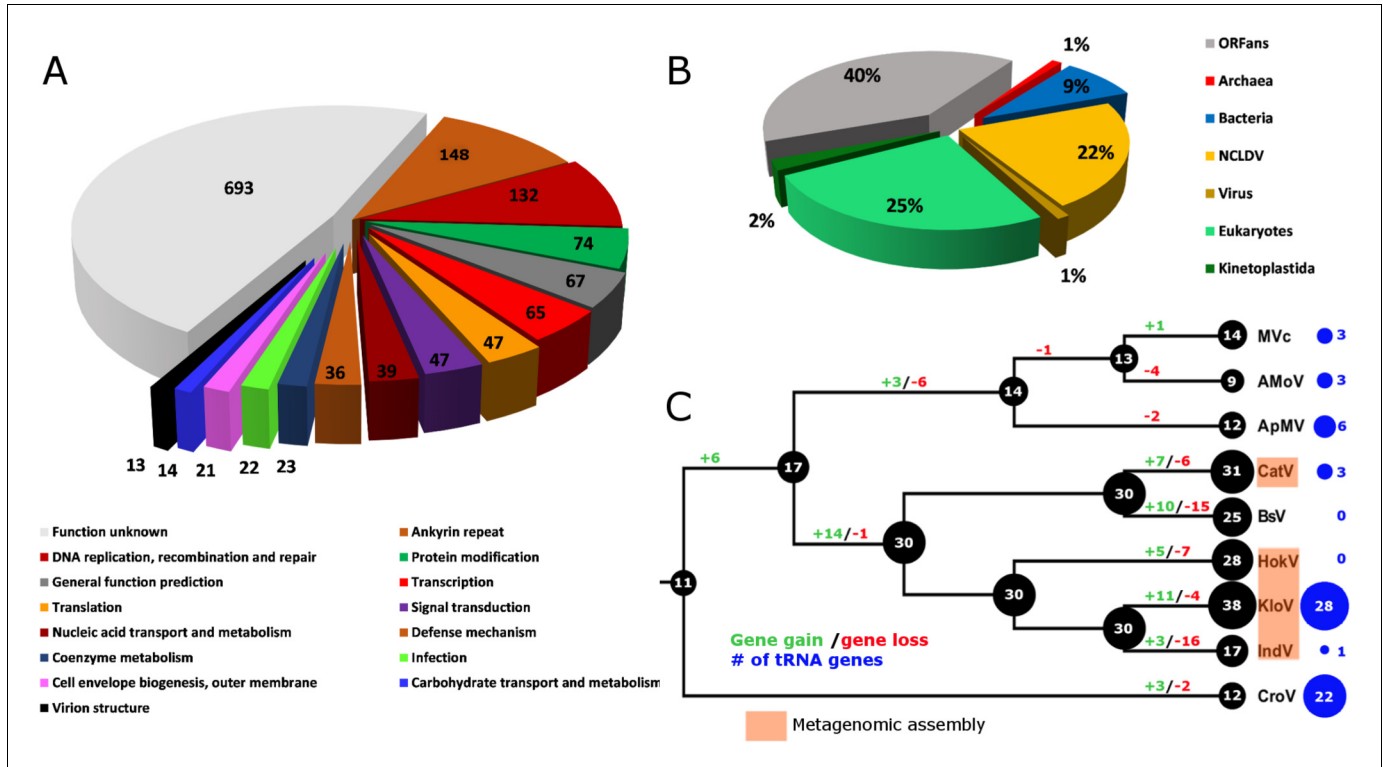

**Figure 4.** BsV genome content. (**A**) Functional assignment of BsV genome content based on BLASTp and CDD rps-BLAST (**B**) Domain of best BLASTp hits (**C**) Evolutionary history of translational machinery found in giant viruses inferred by COUNT. The size of the black circles mapped on a cladogram of the large members of the *Mimiviridae* (see *Figure 6* for full phylogenetic analysis) represents the number of protein coding gene families involved in translation at each node or tip. Blue circles indicate the number of tRNA genes found in each genome. Gene gain and loss events are depicted along the branches. Genomes based on metagenomic assemblies are highlighted to indicate the possibility of incomplete representation of the translation machinery. See *Figure 4—figure supplement 3* for the complete phylogenetic tree for members of the *Mimiviridae*. See *Figure 4—figure supplement 2* for a table of all genes included in the analysis. See *Figure 4—figure supplement 1* for a cladogram depicting the inferred evolutionary history of all gene families in within the *Mimiviridae*. MVc: Megavirus chilensis, AMoV: Acanthamoeba polyphaga Moumouvirus, ApMV: Acanthamoeba polyphage Mimivirus, CatV: Catovirus, BsV: Bodo saltans virus, HokV: Hokovirus, KloV: Klosneuvirus (KlosnV), IndV: Indivirus, CroV: Cafeteria roenbergensis virus.

DOI: https://doi.org/10.7554/eLife.33014.008

The following source data and figure supplements are available for figure 4:

**Source data 1.** Genome content.

DOI: https://doi.org/10.7554/eLife.33014.012

**Figure supplement 1.** Evolutionary history of gene family content found in giant viruses inferred by COUNT.

DOI: https://doi.org/10.7554/eLife.33014.009

**Figure supplement 2.** Comparison of translational machinery encoded by giant viruses based on phylogenetic analysis: BsV Translational machinery compared to other NCLDV: blue: monophyletic group, red: recently host acquired, green: recently acquired from bodonid host, ψ: pseudogene MimV: Mimiviruses, CatV: Catovirus, BsV: Bodo saltans virus, HokV: Hokovirus, KloV: Klosneuvirus (KlosnV), IndV: Indivirus, CroV: Cafeteria roenbergensis virus, PhycV: Phycodnaviridae, PanV: *Pandoraviridae*.

DOI: https://doi.org/10.7554/eLife.33014.010

**Figure supplement 2—source data 1.** Source data for *Figure 4—figure supplement 2*.

DOI: https://doi.org/10.7554/eLife.33014.013

**Figure supplement 3.** Evolutionary history of translational machinery found in giant viruses inferred by COUNT and abundance of ankyrin repeat-domain genes.

DOI: https://doi.org/10.7554/eLife.33014.011

similar to viruses in the nucleocytoplasmic large DNA viruses (NCLDV) group, while the remaining ORFs were most similar to bacterial (9%) or archaeal (1%) sequences. In gene cluster analysis, only 45% of protein-coding gene clusters are shared with related viruses such as CroV and klosneuviruses, highlighting the low number of conserved core genes amongst these viruses (*Figure 4—figure supplement 1*). Essential genes for replication, translation, DNA replication and virion structure are located in the central part of the genome, while the periphery is occupied by duplicated genes, including 148 copies of ankyrin-repeat-containing proteins (*Figure 3*).

## Functional genome content

While no function could be attributed to 54% of ORFs, the largest identifiable fraction of annotations are involved in DNA replication and repair (*Figure 4A*). Coding sequences for proteins associated with all classes of DNA repair mechanisms were identified including DNA mismatch repair (MutS and Uvr helicase/DDEDDh 3′−5′ exonucleases), nucleotide excision repair (family-2 AP endonucleases), damaged-base excision (uracil-DNA glycosylase and formamidopyrimidine-DNA glycosylase) and photoreactivation (deoxyribodipyrimidine photolyase). The repair pathways are completed by DNA polymerase family X and NAD-dependent DNA ligase. Sequences were also found that code for proteins involved in DNA replication, including several primases, helicases, and an intein-containing family-B DNA polymerase, as well as replication factors A and C, a chromosome segregation ATPase, and topoisomerases 1 (two subunits) and 2. Sequences associated with proteins mediating recombination were also identified including endonucleases and resolvases, as well as the aforementioned DNA repair machinery.

There were 47 sequences identified that matched enzymes involved in protein and signal modification, with the majority being serine/threonine kinases/phosphatases. These are potentially involved in host cell takeover.

The genome of BsV is rich in coding sequences involved in transcription. An early transcription factor putatively recognizing the highly conserved AAAAATTGA motif and a late transcription factor putatively targeting TGCG were identified, whereas the target sequence of a third transcription factor is unknown. Further, a TATA-binding protein, a transcription initiation factor (TFIIIB) and a transcription elongation factor (TFIIS) were identified that should aid transcription. As well, RNA polymerase subunits a,b,c,e,f,g and I were identified and are assisted by DNA topoisomerases Type 2 and 1B. BsV encodes amRNA specific RNase III, a poly A polymerase, several 5′ capping enzymes and methyl transferases. Transcription is presumably terminated in a manner similar to that described in ApMV, since hairpin structures were detected in the 3′ UTR of most transcripts (*Priet et al., 2015*). They are probably recognized and processed by the viral encoded RNase III in a manner similar to ApMV (*Byrne et al., 2009*). After hairpin loop cleavage, the poly-A tail is added by the virally encoded poly-A polymerase. The 5′ capping is accomplished by the virus-encoded mRNA capping enzyme, as well as several cap-specific methyltransferases. The extensive cap modification suggests that BsV is independent of the trans-splicing of splice-leader mRNA containing cap structures found in kinetoplastids (*Stuart et al., 1997*).

BsV also encodes several enzymes associated with nucleic-acid transport and metabolism, including several AT-specific nucleic-acid synthesis pathway components. For instance, adenylosuccinate, thymidylate and pseudouridine synthetases and kinases, as well as ribonucleoside-diphosphate reductase were evident. Other ORFs were associated with nucleotide salvaging pathways, including nucleoside kinases, phosphoribosyl transferases, and cytidine and deoxycytidylate deaminase. A mitochondrial carrier protein was identified that, similar to ApMV, likely provides dATP and dTTP directly from the kinetoplast to the virus factory, as evident from electron microscopic observations (*Monné et al., 2007*).

Several genes were identified that are involved in membrane trafficking. A system based on soluble N-ethylmaleimide-sensitive factor (NSF) attachment proteins (SNAPs) and the SNAP receptors (SNAREs) appears to have been acquired from the host by horizontal gene transfer in the recent evolutionary past. In combination with several NSF homologues, including the vesicular-fusion ATPases that also seems to have been acquired from the host. Other proteins putatively involved in membrane trafficking are rab-domain containing proteins, ras-like GTPases, and kinesin motor proteins.

The BsV genome encodes four major capsid proteins. One of these proteins contains several large insertions between conserved domains shared among all four capsid proteins, and with 4194

aa boasts a size almost seven times that of its paralogs. This enlarged version of the major capsid protein might be responsible for creating the halo around the virus particles observed by TEM, by producing shortened fibers similar to those observed in ApMV (*Figure 2A*) (*Xiao et al., 2009*). Further, the genome contains two core proteins, several chaperones and glycosylation enzymes suggesting that proteins are highly modified before being incorporated into the virus particle.

There were numerous ORFs that were similar to genes encoding metabolic proteins, like enzymes putatively involved in carbohydrate metabolism. However, no one continuous metabolic pathway could be assembled and therefore these enzymes likely complement host pathways. BsV also encodes coenzyme synthetases such as CoA and NADH and to meet the demand for amino acids that are rare in the host, BsV encodes the key steps in the synthesis pathways of glutamine, histidine, isoleucine, and asparagine.

Another group of genes putatively mediate competitive interactions, either directly with the host, or with other viruses or intracellular pathogens. These include genes involved in the production of several toxins such as a VIP2-like protein as well as antitoxins containing BRO domains. Further, a partial bleomycin detox pathway was found, as well as multidrug export pumps and partial restriction modification systems.

While BsV encodes a complex translation machinery, it differs markedly from those described in other members of the *Mimiviridae* and shows the largest turnover of these genes resulting in a net contraction (*Figure 4C*). Eukaryotic translation initiation factors include the commonly seen eIF-2a, eIF-2b, eIF-2g, eIF-4A-III and eIF-4E, as well as several pseudogenes related to eIFs. Eukaryotic elongation factor 1 is also present as is eukaryotic peptide chain release factor subunit 1. Notable is the absence of eIF-1; instead, BsV encodes a version of IF-2 that appears to have been acquired from the host and is functionally analogous to eIF-1 in kinetoplastids. The most striking difference to other NCLDVs is the absence of tRNAs. Uniquely among NCLDVs, BsV encodes several tRNA repair genes. These genes include putative RtcB-like RNA-splicing ligase, putative CAA-nucleotidyltransferase, tRNA 2'-phosphotransferase/Ap4A_hydrolase, putative methyltransferase, a TRM13-like protein, pseudouridine synthase and tRNA ligase/uridine kinase. Most of these genes appear to have been recently acquired from the host (*Figure 4—figure supplement 2*). Other translation modification enzymes found in BsV and other NCLDVs include tRNA(Ile)-lysidine synthase, tRNA pseudouridine synthase B and tRNA 2'-phosphotransferase. Similar to the tRNAs, there are few aminoacyl-tRNA synthetases (aaRS) in BsV. Three of the recognizable aaRS are pseudogenes and show signs of recent nonsense mutations or ORF disruptions by genome rearrangements (aspRS, glnRS, and asnRS). The only complete aaRS proteins are isoleucine-tRNA synthetase, found in all members of the *Mimivridae*, and a phenylalanyl-tRNA synthetase.

## Repeat regions

Genes in the genomic periphery have undergone massive duplication, with 148 copies of ankyrin repeat proteins, mostly present in directional tandem orientation (*Figure 3*). These sequences are quite variable and encode between 4 and 17 ankyrin-repeat domains. There is evidence of very recent sequence duplication resulting in direct or inverted repeat regions that contain complete ankyrin-repeat coding sequences and further expand the repeat clusters (*Figure 3—figure supplement 1A*). Interestingly, the 5' coding region of many ankyrin-repeat containing protein ORFs contain fragments of catalytic domains of essential viral genes such as DNA polymerases or the MutS repair protein (*Figure 3—figure supplement 1C*).

## Genomic mobilome

In contrast to described giant virus genomic mobilomes consisting of virophages and transpovirons, the BsV genome is dominated by inteins, autocatalytic proteinases, and self-splicing group 1 introns (*Desnues et al., 2012*; *Fischer and Suttle, 2011*; *Santini et al., 2013*; *La Scola et al., 2008*). These mobile elements spread by targeting the DNA coding regions of essential genes for virus replication by deploying unrelated homing endonucleases encoded by internal ORFs nested within the elements (red ORFs in *Figure 5*). Inteins that are closely related to those in *Mimiviridae* and *Phycodnaviridae* are found in the BsV DNA polymerase family B gene, while other unrelated inteins are found in the DNA-dependent RNA polymerase subunits A and B genes (polr2a and polr2b: *Figure 5*). The inteins in the RNA polymerase genes seem to be devoid of an active homing endonuclease, and are

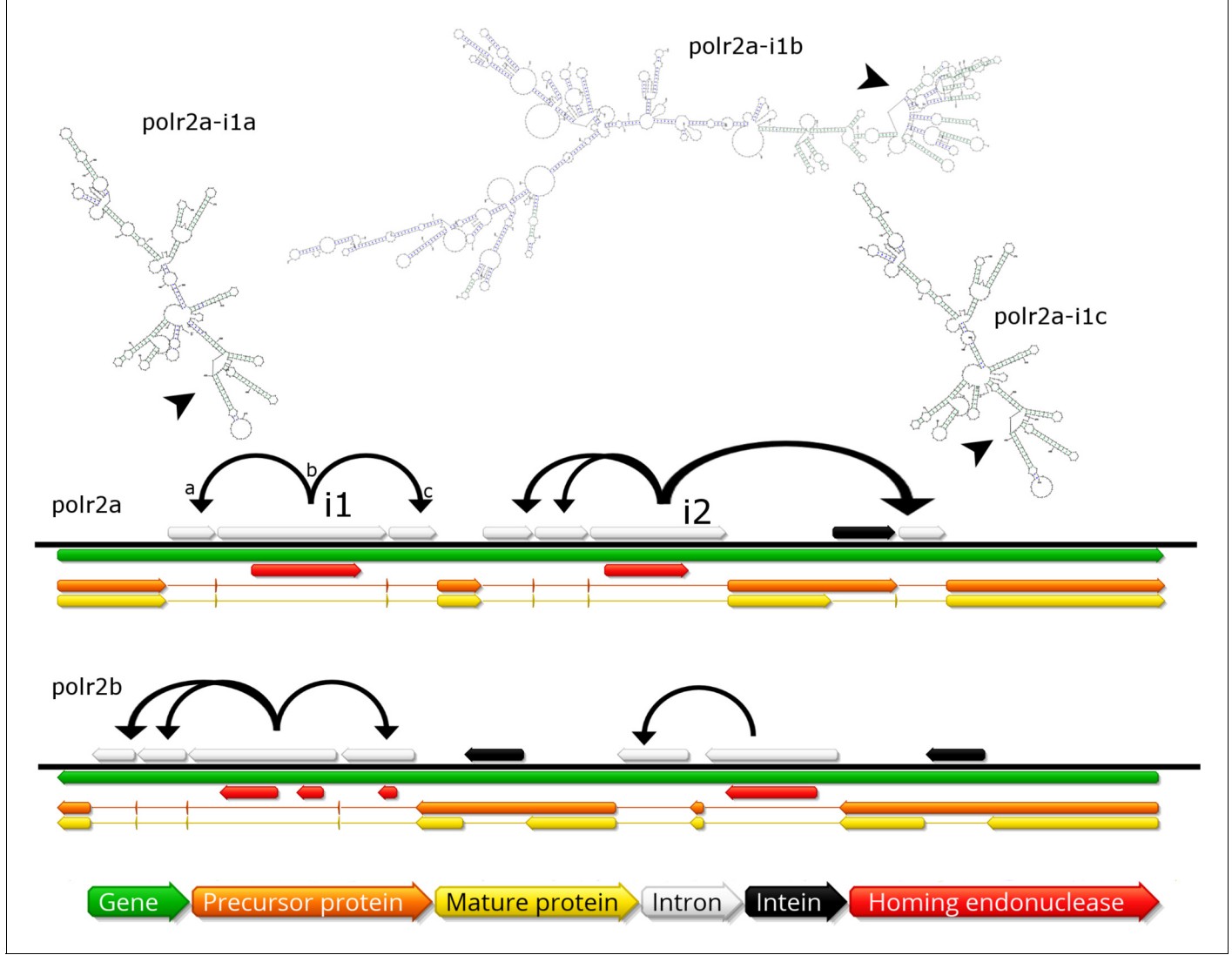

**Figure 5.** Invasion of RNA polymerase genes by selfish genetic elements. Organization of self-splicing group 1 introns (grey) and inteins (black) within genes polr2a and polr2b (green). ORFs within the introns encode homing endonucleases (red). The precursor proteins (orange) are expressed after introns are excised from the pre-mRNA including the internal ORFs encoding the endonucleases. Consequently, the inteins (grey) excise themselves from the precursor protein resulting in the mature protein (yellow). Secondary structure of related self-splicing group 1 introns ('parental' intron polr2a-i1b and 'offspring' polr2a-i1a and –i1c) is shown above the coding sequence with conserved self-splicing catalytic site highlighted by arrows. Additional secondary structure predictions and sequence alignment are available in *Figure 5—figure supplement 1*.

DOI: https://doi.org/10.7554/eLife.33014.014

The following figure supplement is available for figure 5:

**Figure supplement 1.** Self-splicing group 1 introns found in BsV.

DOI: https://doi.org/10.7554/eLife.33014.015

therefore fixed, suggesting an evolutionary ancient invasion. The intein in DNA pol B may be an exception, as an HNH endonuclease is located in close genomic proximity and could promote homing in a trans-acting fashion.

The group 1 self-splicing introns seem to have independently invaded the RNA polymerase subunit 1 and 2 genes, since these introns carry different homing endonucleases (HNH and GIY-YIG type) and their ribozymes differ in secondary structure (*Figure 5*, *Figure 5—figure supplement 1B*). Subsequent to the initial integration of introns containing endonucleases (e.g. polr2a-i1b in *Figure 5*), these homing endonucleases seeded 'offspring' introns within the same gene (e.g. polr2a-i1a and –i1c in *Figure 5*). These secondary introns show conserved secondary RNA structure, but lack the

homing endonuclease of their parental intron. Therefore, the secondary introns probably rely on the trans-homing of their parental introns' endonucleases. The highly conserved sequence for some of the offspring introns (94.4% sequence identity for polr2a-i1a and -i1c, *Figure 5—figure supplement 1A*) suggests that these have spread relatively recently, while other introns that only show conservation in their secondary structure probably represent older invasions. Besides proliferating introns, the BsV genome is also home to two distinct actively proliferating transposon classes.

## Phylogenetic placement and environmental representation of BsV

Phylogenetic analysis of BsV places it within the *Mimiviridae*. Whole genome analysis based on gene cluster presence/absence of NCLDVs resulted in BsV clustering within the large mimiviruses (recently proposed 'Megavirinae') and separate from the small mimiviruses ('Mesomimivirinae') (*Figure 6A*) (*Gallot-Lavallée et al., 2017*). BsV is closest affiliated with the genomes of the Klosneuviruses, assembled from metagenomic data, and to a lesser degree with CroV. Phylogenetic analysis of five concatenated highly conserved NCLDV core genes reproduced this pattern within the *Mimiviridae* (*Figure 6B*). Within the 'Megavirinae', three clades emerged, the *Acanthamoeba*-infecting Mimiviruses, the metagenomic klosneuviruses (Cato-, Hoko-, Klosneu-, and Indivirus) with BsV, and CroV as the sole member of an outgroup (*Figure 6B*). Phylogenies of the individual genes placed BsV within the klosneuviruses in three of five cases (*Figure 6—figure supplements 1,2*). When metagenomic reads of NCLDV DNA polymerase B sequences from the TARA oceans project (http://www.igs.cnrs-mrs.fr/TaraOceans/) were mapped to a maximum likelihood tree of DNA polymerase B sequences fromthe *Mimiviridae*, it was apparent that the 'Megavirinae' were dominated by klosneuviruses, with BsV as their only isolate (*Figure 6C*). Thus, BsV is representative of the largest group of identifiable icosahedral giant viruses in the oceans with CroV being the sole representative of the second most abundant clade.

AaV: *Aureococcus anophagefferens virus*; AcV: *Anomala cumrea entomopoxvirus*; AMaV: *Acanthamoeba castellanii mamavirus*; AMgV: *Moumouvirus goulette*; AMoV: *Acanthamoeba polyphaga moumouvirus*; ApMV: *Acanthamoeba polyphaga mimivirus*; ASFV: *African swine fever virus BA71V*; AtcV: *Acanthocystis turfacea chlorella virus 1*; BpV: *Bathycoccus sp. RCC1105 Virus*; BsV: *Bodo saltans virus NG1*; CatV: Catovirus; CeV: *Chrysochromulina ericina virus 1B*; CroV: *Cafeteria roenbergensis virus BV-PW1*; EhV: *Emiliania huxleyi virus 86*; FauV: *Faustovirus E12*; HokV: *Hokovirus*; HvV: *Heliothis virescens ascovirus 3e*; IiV: *Invertebrate iridescent virus*; IndV: *Indivirus*; ISKV: *Infectious spleen and kidney necrosis virus*; KloV: *Klosneuvirus*; LauV: *Lausannevirus*; MarV: *Marseillevirus T19*; MoVs: *Mollivirus sibericum*; MpVS: *Micromoas pusillae Virus SP-1*; MsV: *Melanoplus sanguinipess entomopoxmvirus*; MVc: *Megavirus chiliensis*; MyxV: *Myxoma virus*; OLV1: *Organic Lake Phycodnavirus 1*; OLV2: *Organic Lake Phycodnavirus 2*; OtV: *Ostreococcus tauri virus 1*; PbCV: *Paramecium bursaria chlorella virus 1*; PgV: *Phaeocystis globosa virus 16T*; PiVs: *Pithovirus sibericum P1084-T*; PoV: *Pyramimonas orientalis virus*; PpV: *Phaeocystis pouchetii virus*; PVd: *Pandoravirus dulcis*; PVs: *Pandoravirus salinus*; SfV: *Spodoptera frugiperda ascovirus 1a*; SGV: *Singapore grouper iridovirus*; TnV: *Trichoplusia ni ascovirus 2 c*; VacV: *Vaccinia virus*; WiV: *Wiseana iridescent virus*; YLV1: *Yellowstone lake phycodnavirus 1*

## Discussion

### BsV represents the most abundant subfamily of the *Mimiviridae* in the TARA oceans data

Particle structure, functional features, like the transcription machinery, and phylogenetic analysis firmly place BsV within the *Mimiviridae*, making it one of the largest completely sequenced genome of the family. BsV groups with the klosneuviruses and is separate from CroV and the *Acanthamoeba* infecting mimiviruses, and falls within the proposed 'Megavirinae' (*Figure 6*). A separate subfamily was proposed for the metagenomic klosneuviruses, the 'Klosneuvirinae', which would make BsV as the first isolate and the type species of this subfamily (*Schulz et al., 2017*). The high representation of the klosneu- and BsV-like viruses in metagenomic reads from the TARA oceans survey is consistent with BsV being representative of the most abundant group of *Mimiviridae*, and possibly all icosahedral giant viruses in the oceans (*Hingamp et al., 2013*). The initial detection of the klosneuviruses in low complexity fresh water metagenomes further supports the global prevalence of this group.

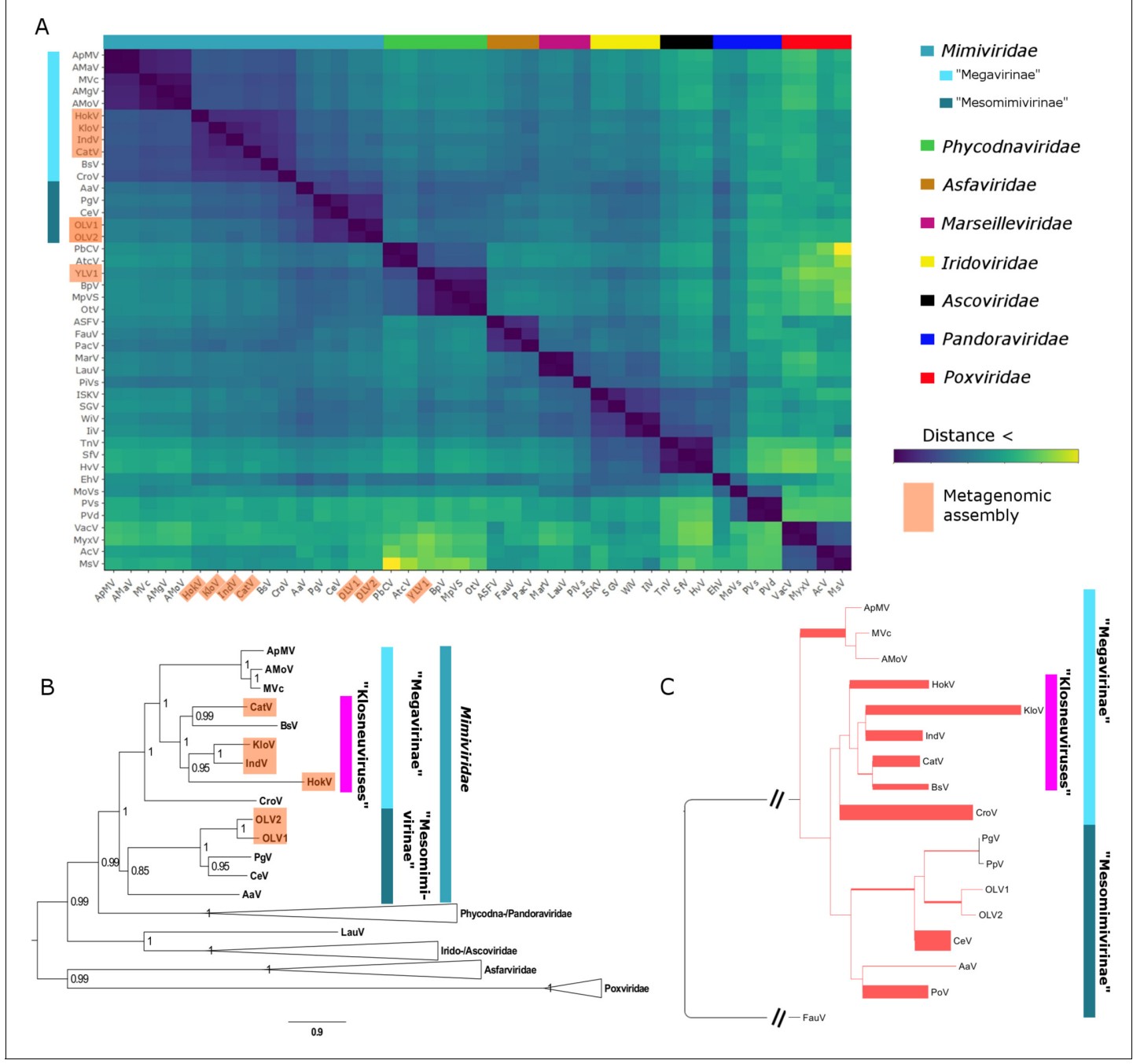

**Figure 6.** BsV Phylogeny (**A**) Phylogenetic distance matrix of NCLDV genomes based on whole genome content of gene family presence/absence. Both axes are identical and 'heat' of the color increases with dissimilarity in genome content. The shaded 'taxa' on the axes indicate viral sequences assembled from metagenomic data. (**B**) Bayesian posterior consensus tree with Bayesian posterior probability of five concatenated Nucleo-Cytoplasmic Virus Orthologous Groups (NCVOGs) from selected NCLDVs based on two independent MCMC chains (16100 generations rel_diff: 0.104001 effsize: 297). The shaded labels at the end of the branches represent 'taxa' based on sequences assembled from metagenomic data (**C**) Maximum likelihood phylogenetic tree of DNA Polymerase family B of BsV within the *Mimiviridae*. Branch width correlated to the distribution of 256 metagenomic sequences identified as NCLDV DNA polymerase Family B genes from the TARA oceans project recruited to the tree with pplacer.

DOI: https://doi.org/10.7554/eLife.33014.016

The following source data and figure supplements are available for figure 6:

**Source data 1.** Phylogenetics.

DOI: https://doi.org/10.7554/eLife.33014.020

**Figure supplement 1.** Phylogenetic analysis of five concatenated NCVOGs from selected NCLDVs.

*Figure 6 continued on next page*

Genomics and Evolutionary Biology | Microbiology and Infectious Disease

*Figure 6 continued*

DOI: https://doi.org/10.7554/eLife.33014.017

**Figure supplement 2.** Phylogenetic analysis of D5-like helicase-primase (NCVOG0023;A-C) and DNA polymerase family B (NCVOG0038;D-F) from selected NCLDVs.

DOI: https://doi.org/10.7554/eLife.33014.018

**Figure supplement 3.** Phylogenetic analysis of DNA or RNA helicases of superfamily II (NCVOG0076; **A–C**), packaging ATPase (NCVOG0249; **D–F**), Poxvirus Late Transcription Factor VLTF3-like (NCVOG0262; **G–I**) from selected NCLDVs.

DOI: https://doi.org/10.7554/eLife.33014.019

## BsV has acquired a host mechanism to facilitate membrane fusion, presumably employed during infection and virion morphogenesis

The SNAP/SNARE membrane fusion system found in BsV appears to have been recently acquired from the bodonid host via horizontal gene transfer. This system could mediate membrane fusion in a pH-dependent manner (*Itakura et al., 2012*). Accordingly, we propose a phagocytosis-based infection strategy for BsV: As described for ApMV, BsV is ingested through the cytostome and is phagocytosed in the cytopharynx before being transported in a phagosome toward the posterior of the cell (*Mutsafi et al., 2010*); here the viral SNAP/SNARE interacts with the host counterparts to initiate the fusion of the inner virus membrane with the phagolysosomal membrane upon phagolysosome acidification, releasing the viral genome into the cytoplasm. This scenario is supported by the localization of the virus factory at the posterior of the cell and virus particle structure (*Figure 2A,C* and *Figure 2—figure supplement 1B–C*). According to this hypothesis, SNAP/SNARE proteins must be present in membranes of the mature virus particles and only get exposed after the stargate opens. The SNAP/SNARE system might also be involved in recruiting membrane vesicles from host organelles to the virus factory during maturation of the virus particle as has been described for pox viruses (*Figure 2D*) (*Laidlaw et al., 1998*).

## The BsV possesses an arsenal of mechanisms that could be involved in interference competition

As a representative of environmentally highly abundant viruses, BsV might regularly experience competition for host resources. The putative toxin-antitoxin systems observed in BsV might be involved in competing with other parasites of viral or prokaryotic nature for these resources, by inhibiting their metabolism or damaging their genome as proposed for ApMV (*Boyer et al., 2011*). Most remarkable, however, are the site-specific homing endonucleases encoded by the self-splicing group 1 introns and inteins that have invaded several genes essential for BsV replication. These invasions also seem to be part of the competitive arsenal of BsV, fending off related virus strains competing for abundant and common hosts such as bodonids. During superinfection of two related viruses, having selfish elements encoding homing endonucleases targeting essential genes might be a competitive advantage. As the two competing virus factories are established in the cytoplasm, the endonucleases encoded within the intron or intein cleave the unoccupied locus in the genome of the intron/intein-free virus. The intron or intein containing virus' genome stays intact, since the target sequences of the endonucleases within its genes are masked by the insertion of the intron or intein. Thus, the intron/intein-containing virus is reducing the ability of the competing virus to replicate (*Figure 7*). A similar mechanism has been described for competing phages in which an intron-encoded or derived homing endonuclease mediates marker exclusion during superinfection, causing selective sweeps of genes in the vicinity of the endonuclease through the phage population (*Belle et al., 2002*; *Goodrich-Blair and Shub, 1996*; *Kutter et al., 1995*). More credence to this hypothesis is given by the RNA polymerase sequences encoded by the proposed catovirus and klosneuvirus (*Schulz et al., 2017*). These genes (polr2a and polr2b) are fragmented in a manner similar to that observed in BsV, and also appear to encode homing endonucleases between gene fragments, suggesting the presence of self-splicing introns is common in the relatives of BsV such as the klosneuviruses. Since the hosts for klosneuviruses are unknown, it is possible that they compete with BsV for the same hosts, or at least experience similar competition. The presence of non-fixed inteins in other giant viruses hints to past invasions and selection for inteins in a manner resembling the hypothesis proposed here for introns (*Culley et al., 2009*; *Gallot-Lavallée et al., 2017*). The retention of fixed

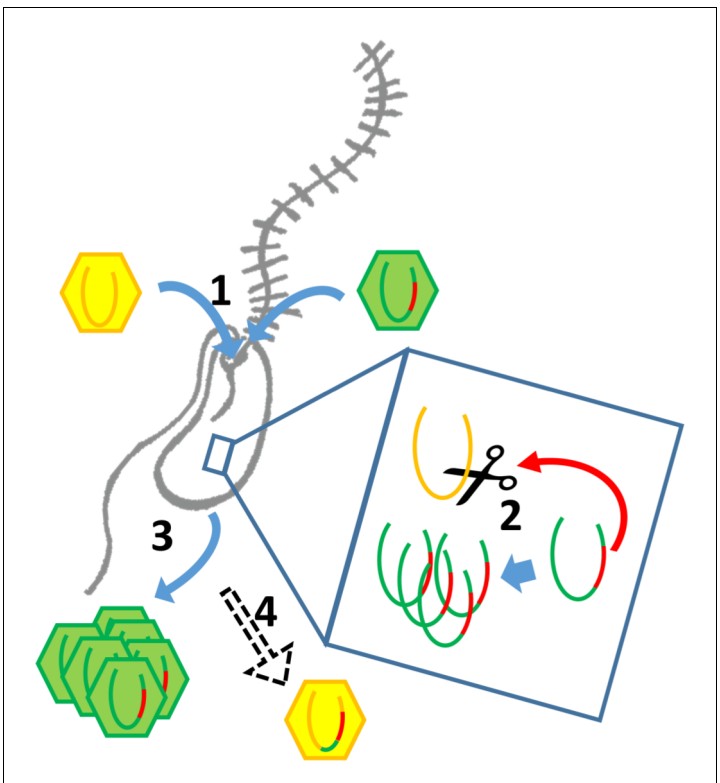

**Figure 7.** Intron/intein encoded endonuclease-mediated interference completion between related viruses. (1) Two related viruses infect the same host cell. The green virus genome contains a selfish element encoding a homing endonuclease. (2) During initial replication, the endonuclease is expressed and cleaves the unoccupied locus on the yellow virus' genome impairing its replication. (3) Due to suppressing its competitor's replication, the majority of the viral progeny is of the green virus' type. (4) The yellow virus can rescue its genome by using the green virus' genome as a template. This creates a chimeric genome containing the selfish element and the endonuclease as well as adjacent sequences originating from the green virus' genome.

DOI: https://doi.org/10.7554/eLife.33014.021

inteins in BsV and other giant viruses suggests that there are still viruses in the environment that encode the relevant endonucleases that apply selective pressure to retain the inteins. A similar situation might explain the presence of an intein in the DNA polymerase of *Pandoravirus salinus*, but not in *P. dulcis*. Thus, pandoraviruses may be an excellent model to experimentally explore the proposed mechanism of intron homing endonuclease mediated competition.

## The contracted translation machinery in BsV suggest that this feature is a homoplasic trait of giant viruses

The absence of tRNAs in the BsV genome is remarkable since tRNAs are found in all complete genomes of giant viruses and even in many moderately sized NCLDV genomes (*Figure 4C*, *Figure 4—figure supplement 3*). This might be an adaptation to the unusual RNA modification found in kinetoplastids that also encompasses tRNA editing (*Alfonzo and Lukeš, 2011*; *Stuart et al., 1997*). Similarly, Trypanosoma mitochondrial tRNAs are exclusively nuclear encoded (*Hancock and Hajduk, 1990*). BsV likely cannot replicate this unusual editing, and thus relies on using host tRNAs. Hence, BsV encodes tRNA repair genes to compensate for the lack of tRNA synthesis and to maintain the available tRNA pool in the host cell. Most of these genes appear to have been recently acquired from the host (*Figure 4C*, *Figure 4—figure supplement 2*). Like the tRNAs, most virus-encoded aminoacyl-tRNA synthetases might not recognize the highly modified tRNAs present in the host and are therefore degrading in the absence of positive selective pressure (*Figure 4C*, *Figure 4—figure supplements 2,3*). The presence of three recognizable pseudogenes of aminoacyl-tRNA synthetases is especially remarkable in this context and argues for an evolutionary recent

degradation (*Figure 4—figure supplement 2*). This turnover in translational machinery components in BsV and the klosneuviruses, combined with the apparent diverse origin of these genes, suggests that the translation machinery found in giant viruses is the result of rapid adaptation by gene acquisition via horizontal gene transfer, as has been recently proposed by *Schulz et al. (2017)*. BsV demonstrates that such genes can be readily purged from the virus genome if they are not required in a new host. Thus, BsV provides further evidence that the translation machinery encoded by NCLDVs is a homoplasic trait and need not be ancient in origin.

## An inflated genomic accordion due to evolutionary arms races is responsible for genome gigantism in BsV and sheds light on giant virus evolution

The 148 copies of ankyrin-repeat domain proteins in the genomic periphery of BsV are telltale signs of an expanded genomic accordion (*Figure 3*) (*Elde et al., 2012*). The observation of almost identical sequences in the very periphery of the genome is consistent with the genomic accordion hypothesis, in which the most recent duplications are closest to the genome ends (*Figure 3*, *Figure 3— figure supplement 1A,B*). The genomic recombinations causing the gene duplications can also lead to the disruption of coding sequences that might explain the comparatively low coding density of BsV. Proteins with ankyrin-repeat domains are multifunctional attachment proteins that in pox viruses determine host range by inhibiting host innate immune system functions (*Camus-Bouclainville et al., 2004*). Further ankyrin-repeat domain proteins are used by bacterial intracellular pathogens like Legionella to manipulate eukaryotic host cells (*Pan et al., 2008*). The presence of fragments of the catalytic domains of essential viral genes in many ankyrin-repeat containing genes is of further importance (*Figure 3—figure supplement 1C*). This suggests a decoy defense mechanism, where these fusion proteins mimic the targets of host antiviral defense systems disrupting essential viral functions. By acting as decoy targets, they immobilize the proposed host factors upon binding via their ankyrin-repeat domains similar to what occurs in vaccinia virus (*Elde et al., 2009*). The immobilized host factors might even be degraded in a ubiquitin-dependent manner reminiscent of the situation in pox viruses as suggested by the presence of several ubiquitin conjugating enzymes encoded in the BsV genome (*Sonnberg et al., 2008*). An ankyrin-repeat-based defense system might explain the observation of cells surviving or avoiding infection that can persist in the presence of the virus (*Figure 1C*). Alternatively, the protein-protein interaction mode of ankyrin repeat proteins might aid attachment and induction of phagocytosis as the bodonid host cells have changing surface antigens (*Jackson et al., 2016*). Whatever the true function of the ankyrin-repeat proteins might be, they clearly highlight the importance of the genomic accordion in giant virus genome evolution driven by evolutionary arms races and complement previous observations of a contracting genomic accordion in ApMV (*Boyer et al., 2011*).

## Summary

Bodo saltans virus (BsV) has the one of the largest sequenced genome of the *Mimiviridae* and is representative of the most abundant members of this family in aquatic ecosystems. BsV is also the first described DNA virus that infects kinetoplasts, or any member of the supergroup Excavata, a major evolutionary lineage of eukaryotes, and is the first isolate of a subfamily within the *Mimiviridae* that was based only on metagenomic data. BsV highlights the genomic plasticity of giant viruses via the genomic accordion, which allows for large-scale genome expansions and contractions via non-homologous recombination. The recent duplications in BsV demonstrate genome expansion in action and exemplifies the mechanisms leading to genome gigantism in the *Mimiviridae*. Further, the function of the expanding genes suggests that strong evolutionary pressure is placed on these viruses by a virus-host arms race that has driven genomic expansion. Genomic plasticity is further apparent in the translational machinery, which shows signs of recent gene loss and rapid adaptation to its bodonid host. This emphasizes that the translational machinery of giant viruses is an acquired homoplasic trait not derived from a common ancestor. An invasion of selfish elements in essential genes suggests interference competition among related viruses for shared hosts. Bodo saltans virus provides significant new insights into giant viruses and their biology.

## Materials and methods

### Sampling and isolation

Virus concentrates were collected from 11 fresh water locations in southern British Columbia, Canada (49°49'4"N, 123° 7'46"W; 49°42'5"N, 123° 8'47"W; 49°37'34"N, 123°12'27"W; 49° 6'12"N, 122° 4'38"W; 49° 5'22"N, 122° 7'1"W; 49°18'10"N, 122°42'9"W; 49° 8'27"N; 123° 3'16"W; 49°13'21"N, 123°12'43"W; 49°13'13"N, 123°12'41"W; 49°14'52"N, 123°13'59"W; 49°15'58"N, 123°15'34"W). To concentrate giant viruses, 20 l water samples were prefiltered with a GF-A filter (Millipore, Bedford, MA; nominal pore size 1.1 um) over a 0.8 um PES membrane (Sterlitech, Kent, WA). Filtrates from all locations were pooled and were concentrated using a 30 kDa MW cut-off tangential flow filtration cartridge (Millipore, Bedford, MA) (*Suttle et al., 1991*).

*Bodo saltans* strain NG, the host of BsV, was isolated from a water sample collected near the sediment surface of the pond in Nitobe Memorial Garden, The University of British Columbia, Canada (49°15'58"N, 123°15'34"W). Clonal cultures were obtained by end-point dilution, and maintained in modified DY-V artificial fresh water media with yeast extract and a wheat grain (*Andersen et al., 2005*). The identity of the host organism was established by 18S sequencing and the strain was deposited at the Canadian Center for the Culture of Microorganisms (http://www3.botany.ubc.ca/cccm/) reference number CCCM 6296 (*von der Heyden and Cavalier-Smith, 2005*). *Bodo saltans* NG cultures were inoculated at approximately $2 \times 10^5$ cells/ml with the pooled giant virus concentrate from all 11 locations. Cell numbers were determined by flow cytometry and compared to a medium only mock-infected control culture (LysoTracker Green (Molecular Probes) vs. FSC on FACS-calibur (Becton-Dickinson, Franklin Lakes, New Jersey, USA))(*Rose et al., 2004*). After a lytic event was observed, the lysate was filtered through a 0.8 um PES membrane (Sterlitech) to remove host cells. The lytic agent was propagated and a monoclonal stock was created by three consecutive end point dilutions. The concentrations of the lytic agent were screened by flow cytometry using SYBR Green (Invitrogen Carlsbad, CA) nucleic acid stain after 2% glutaraldehyde fixation (vs SSC) (*Brussaard, 2004*). Cell numbers represented in *Figure 1C* are supplied in 'Source_data_Fig1C_raw'.

The similarity to the flow cytometry profile of Cafeteria roenbergensis virus suggested that the lytic agent was in deed a giant virus.

### Transmission electron microscopy

#### Negative staining

Bodo saltans lysates after BsV infection were applied to the carbon side of a formvar-carbon coated 400 mesh copper grid (TedPella, CA) and incubated at 4°C in the dark overnight in the presence of high humidity. Next, the lysate was removed and the grids were stained with 1% Uranyl acetate for 30 s before observation on a Hitachi H7600 transmission electron microscope at 80 kV.

#### High-pressure freezing and ultra-thin sectioning

Exponentially growing *B. saltans* cultures were infected at a concentration of $5 \times 10^5$ cells ml$^{-1}$ with BsV at a relative particle to cell ratio of ~5 to ensure synchronous infection. Cells were harvested from infected cultures at different time points (6, 12, 18, 24 hr post-infection) as well as from uninfected control cultures. Cells from 50 ml were pelleted in two consecutive 10 min at 5000 xg centrifugation runs in a Beckmann tabletop centrifuge. Pellets were resuspended in 10–15 μl DY-V culture medium with 20% (w/v) BSA and immediately place on ice. Cell suspensions were cryo-preserved using a Leica EM HPM100 high-pressure freezer. Vitrified samples were freeze-substituted in a Leica AFS system for 2 days at −85°C in a 0.5% glutaraldehyde 0.1% tannic acid solution in acetone, then rinsed 10 times in 100% acetone at −85°C, and transferred to 1% osmium tetroxide, 0.1% uranyl acetate in acetone and stored for an additional 2 days at −85°C. The samples were then warmed to −20°C over 10 hr, held at −20°C for 6 hr to facilitate osmication, and then warmed to 4°C over 12 hr. The samples were then rinsed in 100% acetone 3X at room temperature and gradually infiltrated with an equal part mixture of Spurr's and Gembed embedding media. Samples were polymerized in a 60°C oven overnight. 50 nm thin sections were prepared using a Diatome ultra 45° knife (Diatome, Switzerland) on an ultra-microtome. The sections were collected on a 40x copper grid and stained for 10 min in 2% aqueous uranyl acetate and 5 min in Reynold's lead citrate. Image data were recorded on a Hitachi H7600 transmission electron microscope at 80 kV. Image J (RRID:SCR_

003070) was used to compile all TEM images. Adjustments to contrast and brightness levels were applied equally to all parts of the image.

## Virus concentration and sequencing

For Illumina sequencing, exponentially growing *B. saltans* cultures were infected at a concentration of approximately $5 \times 10^5$ cells ml$^{-1}$ with BsV lysate ($10^7$ VLP ml$^{-1}$) at a multiplicity of infection (MOI) of ~0.5. After 4 days, when host cell densities had dropped below 30%, cultures were centrifuged in a Sorvall SLC-6000 for 20 min, 5000 rpm, 4°C to remove remaining host cells and the supernatant was consecutively subjected to tangential flow filtration with at 30 kDa cut-off (Vivaflow PES) and concentrated approximately 100x. Viral concentrates were subjected to ultracentrifugation at 28,000 rpm, 15°C for 8 hr in a Ti90 fixed angle rotor (Beckman-Coulter, Brea, CA). Pellets were resuspended and virions lysed using laurosyl acid and proteinase K subjected to pulsed-field gel electrophoresis on a CHEF II pulse field gel electrophoresis aperture (BioRad) for 25 hr at 14°C in a 0.8% LMP agarose gel with 60–180S switchtimes and 16.170 ramping factor in 0.5 TBE under 5.5 V/cm and 120°. Genomic DNA was visualized under UV light after 30 min SYBR gold (Invitrogen Carlsbad, CA) staining. The dominant PFGE band belonging to genomic BsV DNA (1.35 Mb) was cut and DNA was extracted using a GELase kit (Illumina, San Diego) and ethanol purified according to manufacturer's protocol. Libraries were prepared using the Illumina Nextera XT kit (Illumina, San Diego, CA) as per manufacturer's recommendation and library quality and quantity were checked by Bioanalyzer 2100 with the HS DNA kit (Agilent Technology). 300 bp paired-end sequencing was performed on an Illumina MiSeq platform by UCLA's Genoseq center (Los Angeles, CA) to a nominal sequencing depth of 800x. Sequence quality was examined using FastQC (RRID:SCR_014583, http://www.bioinformatics.bbsrc.ac.uk/projects/fastqc/) and sequence reads were quality trimmed (parameters: minlen = 50 qtrim = rl trimq = 15 ktrim = r k = 21 mink = 8 ref=$adapters hdist = 2 hdist2 = 1 tbo = t tpe = t) and cleared of human (parameters: minid = 0.95 maxindel = 3 bwr = 0.16 bw = 12 quickmatch fast minhits = 2 qtrim = lr trimq = 10) and PhiX (parameters: k = 31 hdist = 1) sequences against the whole respective genomes using BBMap v35 (http://sourceforge.net/projects/bbmap/).

For PacBio sequencing, BsV was concentrated using precentrifugation and TFF analogously to the Illumina sequencing step. Next, the concentrate was further concentrated by sedimenting it onto a 40% Optiprep 50 mM Tris-Cl, pH 8.0, 2 mM MgCl2 cushion for 30 min at 28,000 rpm, 15°C in a SW40Ti rotor in an ultracentrifuge (Beckman-Coulter, Brea, CA). An Optiprep (Sigma) gradient was created by underlaying a 10% Optiprep solution in 50 mM Tris-Cl, pH 8.0, 2 mM MgCl2 with a 30% solution followed by a 50% solution and was equilibration over night at 4°C. One ml of viral concentrate from the 40% cushion was added atop the gradient and the concentrate was fractionated by ultracentrifugation in an SW40 rotor for 4 hr at 25,000 rpm and 18°C. The viral fraction was extracted from the gradient with a syringe and washed twice with 50 mM Tris-Cl, pH 8.0, 2 mM MgCl$_2$ followed by centrifugation in an SW40 rotor for 20 min at 7200 rpm and 18°C and were finally collected by centrifugation in an SW40 rotor for 30 min at 7800 rpm and 18°C. Purity of the concentrate was verified by flow cytometry (SYBR Green (Invitrogen Carlsbad, CA) vs SSC on a FACScalibur (Becton-Dickinson, Franklin Lakes, NJ). High-molecular-weight genomic DNA was extracted using phenol-chloroform-chloroform extraction. Length and purity were confirmed by gel electrophoresis and Bioanalyzer 2100 wit the HS DNA kit (Agilent Technology). PacBio RSII 20 kb sequencing was performed by the sequencing center of the University of Delaware. Reads were assembled using PacBio HGAP3 software with 20 kb seed reads and resulted in a single viral contig of 1,384,624 bp, 286.1x coverage, 99.99% called bases and a consensus concordance of 99.9551% (*Chin et al., 2013*).

Cleaned up Illumina reads were mapped to the PacBio contig to confirm the PacBio assembly as well as extending the contig's 5' end by 1245 bp to a total viral genome length of 1,385,869 bp.

## Annotation

Open-reading frames were predicted using GLIMMER (RRID:SCR_011931, *Delcher et al., 2007*) with a custom start codon frequency of ATG, GTG, TTG, ATA, ATT at 0.8,0.05,0.05,0.05,0.05 as well as stop codons TAG, TGA, TAA, minimum length 100 bp, max overlap 25, max threshold 30.

Promoter motives were analyzed by screening the 100 bp upstream region of CDS using MEME (RRID:SCR_001783, *Bailey et al., 2009*). tRNAs were predicted with tRNAscanSE (RRID:SCR_010835, *Lowe and Eddy, 1997*). Group 1 introns were predicted by disruptions in coding sequences

and secondary RNA structure was predicted using S-fold (*Ding et al., 2004*). Intron splicing was confirmed using RT-PCR and Sanger sequencing with gene-specific primers designed to span the predicted splice sited predicted by S-fold. Functional analysis of CDS was performed after translation with BLASTp against the nr database with an e-value threshold of $10^{-5}$ as well as rps conserved domain search against CDD v3.15. Coding sequences were manually assigned to functional classes based on predicted gene function using Geneious R9 (*Kearse et al., 2012*).

The annotated genome of BsV-NG1 was deposited in GenBank under the accession number MF782455. The *Bodo saltans* NG 18S sequence was deposited in GenBank under the accession number MF962814.

## Phylogenetics

Whole genome content phylogeny was performed by OrthoMCL (RRID:SCR_007839, *Li et al., 2003*). Available whole genome sequences of NCLDV from NCBI were downloaded. We first performed gene clustering using OrthoMCL (42) with standard parameters (Blast E-value cutoff = $10^{-5}$ and mcl inflation factor = 1.5) on all protein-coding genes of length ≥100 aa. This resulted in the definition of 3001 distinct clusters. Gene clusters are available in source file 'Source_data_Fig4_-S3_all-Mimi_COGs', 'Source_data_Fig4C_translation-Mimi_COGs', 'Source_data_Fig6A-Fig4C_orthoMCL_groups', and 'Source_data_Fig6A_gene-clusters'. We computed a presence/absence matrix based on the genes clusters and calculated a distance matrix using the according to *Yutin et al. (2009)* (*Yutin et al., 2009*). The R script is provided in 'Source_data_Fig6A_script'. Gene clusters were used to infer ancestral gene content by posterior probabilities in a phylogenetic birth-and-death model in COUNT (*Csurös, 2010*) The COUNT file is available under Source_data_Fig4c_-count-session. Additional gene substitutions were added to the model where phylogenetic analysis of individual genes strongly suggested accordingly (see next section). The Alignments and phylogenetic tees can be found as 'Source_data_Fig4_S1-2_XX_YY', where 'XX' stands for the respective gene and 'YY' stands for 'aln' or 'tree' designates an alignment or tree file.

Alignments of aa sequences were performed in Geneious R9 (RRID:SCR_010519) using MUSCLE with default parameters (RRID:SCR_011812, *Edgar, 2004*). Proteins used for the concatenated NCVOG tree were DNA polymerase elongation subunit family B (NCVOG0038), D5-like helicase-primase (NCVOG0023), packaging ATPase (NCVOG0249), Poxvirus Late Transcription Factor VLTF3-like (NCVOG0262), and DNA or RNA helicases of superfamily II (NCVOG0076) (*Yutin et al., 2014*). Residues not present in at least 2/3 of the sequences were trimmed and ProtTest 3.2 was used for amino-acid substitution model selection (RRID:SCR_014628, *Darriba et al., 2011*). The resulting alignment can be found under 'Source_data_Fig6B_ncvog_cat' as well as 'Source_data_Fig6-S_ncvogXXXX' representing alignments for the individual gene trees, where 'XXXX' represents the NCVOG number. Maximum likelihood trees were constructed with RAxML rapid bootstrapping and ML search with 1000 Bootstraps utilizing the best fitting substitution matrixes determined by prottest (RRID:SCR_006086, *Stamatakis, 2014*). Maximum likelihood trees of translational genes, based on alignments by *Schulz et al. (2017)* where available, were constructed using PhyML (RRID:SCR_014629, *Guindon et al., 2010*; *Schulz et al., 2017*). Phylogenetic trees were computed with PhyloBayes-7MPI 1.4 f in two Markov Chain Monte Carlo chains under the CAT-GTR model for 10,000 to 25,000 generations. The consensus tree was based on both chains, removing the first 1000 generations. Convergence was confirmed with bpcomp and tracecomp (RRID:SCR_006402, *Lartillot et al., 2013*). Trees were visualized in Figtree (A. Rambaut - http://tree.bio.ed.ac.uk/software/figtree/).

Translated environmental assemblies identified by Hingamp et al. as representing NCLDV DNA polymerase B family genes (http://www.igs.cnrs-mrs.fr/TaraOceans/) were mapped to a *Mimiviridae* DNA polymerase B family reference tree created as described above with pplacer (*Hingamp et al., 2013*; RRID:SCR_004737, *Matsen et al., 2010*). Environmental reads were aligned to the reference alignment using clustalw and were mapped under Bayesian setting. The fat tree was visualized with Archaeopteryx (https://sites.google.com/site/cmzmasek/home/software/archaeopteryx). Of the 401 input sequences, 256 mapped within the *Mimiviridae* and are displayed in the figure.

## Acknowledgements

We thank members of the Suttle Lab, past and present, and especially Andrew Lang and Matthias Fischer for helpful comments. As well, the assistance of Jan Finke, Marli Vlok, Marie-Claire Veilleux-

Foppiano and Amy M Chan is greatly appreciated, who directly assisted in sample collection, laboratory work, and ensuring the equipment and supplies were available. Further, we would like to thank the following individuals for their contributions: Grieg Steward for insights into sample preparation and concentration. Denis Tikhonenkov for providing *Bodo saltans* HFCC12. John Archibald and Julius Lukeš for valuable feedback on bodonid biology. Thor Veen for assistance with R script writing. The UBC Bioimaging Facility for feedback and assistance with electron microscopy. Craig Kornak for his protist sketches. The hosts from the podcast 'This Week in Virology' for their valuable comments on the preprint. The work was supported by grants to Curtis A Suttle from the Natural Sciences and Engineering Research Council of Canada (NSERC; 05896), Canada Foundation for Innovation (25412), British Columbia Knowledge Development Fund, and the Canadian Institute for Advanced Research (IMB). Christoph Deeg was supported in part by a fellowship from the German Academic Exchange Service (DAAD), and Cheryl ET Chow by a Centre for Microbial Diversity and Evolution postdoctoral scholarship funded through the Tula Foundation.

## Additional information

### Funding

| Funder | Grant reference number | Author |
| --- | --- | --- |
| Natural Sciences and Engineering Research Council of Canada | 05896 | Curtis A Suttle |
| Canada Foundation for Innovation | 25412 | Curtis A Suttle |
| British Columbia Knowledge Development Fund | | Curtis A Suttle |
| Canadian Institute for Advanced Research | | Curtis A Suttle |
| German Academic Exchange Service | | Christoph M Deeg |
| Tula Foundation | | Cheryl-Emiliane T Chow<br>Curtis A Suttle |

The funders had no role in study design, data collection and interpretation, or the decision to submit the work for publication.

### Author contributions

Christoph M Deeg, Conceptualization, Data curation, Software, Formal analysis, Validation, Investigation, Visualization, Methodology, Writing—original draft, Writing—review and editing; Cheryl-Emiliane T Chow, Supervision, Investigation; Curtis A Suttle, Conceptualization, Supervision, Funding acquisition, Writing—review and editing

### Author ORCIDs

Christoph M Deeg http://orcid.org/0000-0002-4459-9372
Curtis A Suttle http://orcid.org/0000-0002-0372-0033

### Decision letter and Author response

Decision letter https://doi.org/10.7554/eLife.33014.034
Author response https://doi.org/10.7554/eLife.33014.035

## Additional files

### Supplementary files

• Transparent reporting form
DOI: https://doi.org/10.7554/eLife.33014.022

## Major datasets

The following datasets were generated:

| Author(s) | Year | Dataset title | Dataset URL | Database, license, and accessibility information |
|---|---|---|---|---|
| Deeg CM, Suttle CA, Chow C-ET | 2017 | Bodo saltans virus NG1, complete genome | https://www.ncbi.nlm.nih.gov/nuccore/MF782455.1 | Publicly available at the NCBI GenBank (accession no. MF782455) |
| Deeg CM, Suttle CA, Chow C-ET | 2017 | Bodo saltans isolate NG 18S ribosomal RNA gene, partial sequence | https://www.ncbi.nlm.nih.gov/nuccore/MF962814.1 | Publicly available at the NCBI GenBank (accession no. MF962814) |

The following previously published datasets were used:

| Author(s) | Year | Dataset title | Dataset URL | Database, license, and accessibility information |
|---|---|---|---|---|
| Schulz F, Yutin N, Ivanova NN, Ortega DR, Lee TK, Vierheilig J, Daims H, Horn M, Wagner M, Jensen GJ, Kyrpides NC, Koonin EV, Woyke T | 2017 | Giant viruses with an extended complement of translation system components | https://bitbucket.org/berkeleylab/klosneuvirus | Publicly available at bitbucket.org (https://bitbucket.org) |
| Hingamp P, Grimsley N, Acinas SG, Clerissi C, Subirana L, Poulain J, Ferrera I, Sarmento H, Villar E, Lima-Mendez G, Faust K, Sunagawa S, Claverie JM, Moreau H, Desdevises Y, Bork P, Raes J, de Vargas C, Karsenti E, Kandels-Lewis S, Jaillon O, Not F, Pesant S, Wincker P, Ogata H | 2012 | Shotgun Sequencing of Tara Oceans DNA samples corresponding to size fractions for large DNA viruses | https://www.ebi.ac.uk/ena/data/view/ERA155562 | Publicly available at the NCBI GenBank (accession no. ERA155562) |
| Hingamp P, Grimsley N, Acinas SG, Clerissi C, Subirana L, Poulain J, Ferrera I, Sarmento H, Villar E, Lima-Mendez G, Faust K, Sunagawa S, Claverie JM, Moreau H, Desdevises Y, Bork P, Raes J, de Vargas C, Karsenti E, Kandels-Lewis S, Jaillon O, Not F, Pesant S, Wincker P, Ogata H | 2012 | Shotgun Sequencing of Tara Oceans DNA samples corresponding to size fractions for prokaryotes. | https://www.ebi.ac.uk/ena/data/view/ERA155563 | Publicly available at the NCBI GenBank (accession no. ERA155563) |

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
