## [Decision Letter]

Thank you for submitting your article "The kinetoplastid-infecting Bodo saltans virus (BsV), a window into the most abundant giant viruses in the sea" for consideration by *eLife*. Your article has been reviewed by three peer reviewers, and the evaluation has been overseen by a Reviewing Editor and Diethard Tautz as the Senior Editor. The following individuals involved in review of your submission have agreed to reveal their identity: James Van Etten (Reviewer #1); Eugene Koonin (Reviewer #2); Grieg Steward (Reviewer #3).

The reviewers have discussed the reviews with one another and the Reviewing Editor has drafted this decision to help you prepare a revised submission.

The reviewers all agree that this manuscript and work are important but feel that the inferences about its exact phylogenetic position are not sufficiently supported to justify some of the statements and that there are many small editorial fixes needed as found below in the detailed reviews.

Essential revisions:

1) The uncertainty with the phylogenies seems to be the main problematic issue with this manuscript. This can be resolved by simply making various statements about the abundance and phylogeny more cautious and some additional phylogeny analysis as described in reviews 2 and 3.

2) Addressing the editorial revisions requested in the detailed reviews below.

*Reviewer #1:*

This manuscript describes the isolation and characterization of a giant virus (BsV) that infects the ecologically important microzooplankton, the kinetoplastid Bodo saltans. The manuscript is important for several reasons including: i) BsV has the largest genome of viruses included in the *Mimiviridae* family, ii) BsV is the first virus to be isolated and characterized in one of the 3 subclades of the *Mimiviridae* (the other members in this subclade only exist from metagenomic sequences), iii) this subclade of viruses appear to be the most common viruses in the ocean after bacteriophage, iv) BsV is claimed to be one of the few *Mimiviridae* members that does not infect amoeba (note, even though the authors make this claim, it is not clear if they actually tried to infect an amoeba), and v) the genome has some interesting features, including the absence of tRNA genes, it has several homing endonuclease-encoding self-splicing introns and many apparent gene duplications of ankyrin repeat encoding genes.

The experiments appear to be carefully conducted by an established research group that works in the area of giant viruses and viruses in the marine environment. I only have a few comments to make about the manuscript.

1) The authors claim that BsV has a stargate structure, which has been extensively studied in the prototype Mimivirus APMV. In the last paragraph of the subsection “Virus morphology and Replication Kinetics”, they refer to Figure 2 and Figure 2—figure supplement 1. This seems like a very awkward way to refer to a supplemental figure. Figure This awkward notation appears throughout the text). More importantly, the stargate structure is not immediately obvious to me in the two figures cited. Maybe they have better images to illustrate the stargate structure. This is important for another reason in that in the aforementioned paragraph, they say that BsV packages its DNA in the particle at the vertex on the opposite side of the stargate structure. However, it is not obvious to me how they draw that conclusion from Figure 2.

2) The authors state in the first paragraph of the subsection “Virus morphology and Replication Kinetic”, that the virion has 6 layers that engulf the virion and refer to Figure 2. The authors should use arrows or arrow heads to point these out.

*Reviewer #2:*

The isolation and characterization of a novel, highly abundant giant virus of the family *Mimiviridae* from a marine protist is of major interest and value. It is particularly important that the actual virus has been obtained given that the related Klosneuviruses so far have been identified only from metagenomic sequences. This report will be of interest to all virologists, and I believe, also to a broader audience of biologists.

I do have certain concerns regarding the genome analysis described in the manuscript and the authors' interpretation.

My first and probably most important problem is with the phylogeny of the NCLDV shown in Figure 6. I find the tree topologies to be rather suspicious. In the tree of 5 concatenated NCVOGs (6C), BsV forms a long branch, and there is only 65% bootstrap support for the "*Aquavirinae*". The DNAP tree (6D) does not agree, here BsV falls within the Klosneuvirus group. In this situation, I think there is little confidence in the exact position of BsV within the *Mimiviridae*, and also in the position of CroV at the base of the "*Aquavirinae*" as opposed to outside of the MImi-Klos clade as observed in the Schul et al. paper on Klosneuviruses. I find it desirable to invest some more effort into phylogenetic analysis, in particular, analyze Bayesian trees, perform tree topology tests (e.g. AU) and present trees for individual core NCLDV genes. Unless such additional analyses substantially increase the confidence in the tree topology (and I am not entirely optimistic), I think it is best to be much more circumspect regarding the phylogeny of the *Mimiviridae* in general and the position of BsV, in particular.

Second, and related to my first concern, the tree topology affects the conclusions regarding gene gain and loss. The authors indicate in the Abstract and other parts of the article that much of the translation machinery has been lost by BsV. Qualitatively, this is likely to be true but how much, critically depends on the tree topology. Furthermore, I find it highly desirable to perform a formal reconstruction of gene gain and loss using COUNT or GLUM software. Without such an analysis, claims of gene gain and loss remain poorly substantiated. Finally, on the subject of gene loss, to me, one of the most interesting findings here is the presence in BsV of pseudogenes of 3 aaRS pseudogenes and one translation factor. This finding does indeed imply that the translation system could be "on its way out", and in my opinion, should be emphasized stronger.

Third, and less important, I find it strange that the authors claim that BsV has the largest number of genes among the *Mimiviridae*, 1227. Two of the Klosneuviruses are bigger.

Fourth, and least important, but still essential, the authors should be more cautious with taxonomy, especially, given the aforementioned uncertainties with the phylogeny. It is fine to propose Aquavirinae but it should not be 'a distinct evolutionary group [what's that?] *Aqauvirinae*' but rather 'a provisional subfamily "Aquavirinae" with the family *Mimiviridae*'.

*Reviewer #3:*

This paper describes a new viral isolate within the *Megaviridae* and its genomic content. The work is noteworthy for several reasons: 1) the size of the viral genome, which is the largest yet described for an isolate within this family, 2) the unusual functional features of the genome relative to other members of the family, and 3) the lineage of the host, since it is the first DNA virus known to infect a member of the Excavata. This is a noteworthy discovery with lots of interesting tidbits including the absence of tRNAs and the presence of parasitic genetic elements (inteins and introns) that would seem to confer competitive advantage during co-infection with other viruses. The science looks sound. My main general recommendation is that the authors be more careful with the wording, specifically properly qualifying statements about the abundance and phylogenetic position of this virus and its relatives. Also, I would recommend being more circumspect in the phrasing of the headings in the Discussion (and perhaps title), many of which read as statements of fact but are inferences and hypotheses.

---

## [Author Response]

Reviewer #1:[…] 1) The authors claim that BsV has a stargate structure, which has been extensively studied in the prototype Mimivirus APMV. In the last paragraph of the subsection “Virus morphology and Replication Kinetics”, they refer to Figure 2 and Figure 2—figure supplement 1. This seems like a very awkward way to refer to a supplemental figure. This awkward notation appears throughout the text). More importantly, the stargate structure is not immediately obvious to me in the two figures cited. Maybe they have better images to illustrate the stargate structure. This is important for another reason in that in the aforementioned paragraph, they say that BsV packages its DNA in the particle at the vertex on the opposite side of the stargate structure. However, it is not obvious to me how they draw that conclusion from Figure 2.

Supplemental figure references are formatted according to *eLife* guidelines. We agree that the evidence of a stargate structure in the figures is not definitive, but it is persuasive, especially in the figures in the supplemental material. We have modified the statement in the last paragraph of the subsection “Virus morphology and Replication Kinetics” to emphasize uncertainty in our statement.

2) The authors state in the first paragraph of the subsection “Virus morphology and Replication Kinetic”, that the virion has 6 layers that engulf the virion and refer to Figure 2. The authors should use arrows or arrow heads to point these out.

Arrow heads have been added to indicate these layers.

Reviewer #2:[…] I do have certain concerns regarding the genome analysis described in the manuscript and the authors' interpretation.My first and probably most important problem is with the phylogeny of the NCLDV shown in Figure 6. I find the tree topologies to be rather suspicious. In the tree of 5 concatenated NCVOGs (6C), BsV forms a long branch, and there is only 65% bootstrap support for the "Aquavirinae". The DNAP tree (6D) does not agree, here BsV falls within the Klosneuvirus group. In this situation, I think there is little confidence in the exact position of BsV within the Mimiviridae, and also in the position of CroV at the base of the "Aquavirinae" as opposed to outside of the MImi-Klos clade as observed in the Schul et al. paper on Klosneuviruses. I find it desirable to invest some more effort into phylogenetic analysis, in particular, analyze Bayesian trees, perform tree topology tests (e.g. AU) and present trees for individual core NCLDV genes. Unless such additional analyses substantially increase the confidence in the tree topology (and I am not entirely optimistic), I think it is best to be much more circumspect regarding the phylogeny of the Mimiviridae in general and the position of BsV, in particular.

Additional phylogenetic analysis was performed according to the suggestions of reviewer 2. A refined concatenated alignment of five conserved genes followed by Bayesion analysis confidently placed BsV within the klosneuvirus clade. Figure 6 (including figure supplements) and the manuscript have been updated to represent this updated phylogenetic placement.

Second, and related to my first concern, the tree topology affects the conclusions regarding gene gain and loss. The authors indicate in the Abstract and other parts of the article that much of the translation machinery has been lost by BsV. Qualitatively, this is likely to be true but how much, critically depends on the tree topology. Furthermore, I find it highly desirable to perform a formal reconstruction of gene gain and loss using COUNT or GLUM software. Without such an analysis, claims of gene gain and loss remain poorly substantiated. Finally, on the subject of gene loss, to me, one of the most interesting findings here is the presence in BsV of pseudogenes of 3 aaRS pseudogenes and one translation factor. This finding does indeed imply that the translation system could be "on its way out", and in my opinion, should be emphasized stronger.

COUNT analysis of conserved gene families was performed. The reconstructed evolutionary history of translational genes is now depicted in Figure 4 (as well as in the figure supplements) and supports the claims of gene turnover made in the manuscript. The discussion and implication of this turnover have been expanded in the manuscript.

Third, and less important, I find it strange that the authors claim that BsV has the largest number of genes among the Mimiviridae, 1227. Two of the Klosneuviruses are bigger.

The manuscript has been modified to indicate that we are referring to completely sequenced genomes from cultured viruses. This is in contrast to the larger, incomplete klosneuvirus genomes inferred from metagenomic analysis.

Fourth, and least important, but still essential, the authors should be more cautious with taxonomy, especially, given the aforementioned uncertainties with the phylogeny. It is fine to propose Aquavirinae but it should not be 'a distinct evolutionary group [what's that?] Aqauvirinae' but rather 'a provisional subfamily "Aquavirinae" with the family Mimiviridae'.

Based on the updated phylogenetic analysis, the term “*Aquavirinae*” was dropped from the manuscript. The use of taxonomic names previously proposed, but not confirmed by the ICTV (International Committee on Taxonomy of Viruses) throughout the manuscript has been updated as per the reviewer’s suggestion.

Reviewer #3:This paper describes a new viral isolate within the Megaviridae and its genomic content. The work is noteworthy for several reasons: 1) the size of the viral genome, which is the largest yet described for an isolate within this family, 2) the unusual functional features of the genome relative to other members of the family, and 3) the lineage of the host, since it is the first DNA virus known to infect a member of the Excavata. This is a noteworthy discovery with lots of interesting tidbits including the absence of tRNAs and the presence of parasitic genetic elements (inteins and introns) that would seem to confer competitive advantage during co-infection with other viruses. The science looks sound. My main general recommendation is that the authors be more careful with the wording, specifically properly qualifying statements about the abundance and phylogenetic position of this virus and its relatives. Also, I would recommend being more circumspect in the phrasing of the headings in the Discussion (and perhaps title), many of which read as statements of fact but are inferences and hypotheses.

Statements have been modified throughout the manuscript as per the reviewer’s recommendations.